# Concomitant and Bismuth Quadruple Therapy for *Helicobacter pylori* Eradication in Southern Italy: Preliminary Data from a Randomized Clinical Trial

**DOI:** 10.3390/antibiotics13040348

**Published:** 2024-04-10

**Authors:** Giuseppe Losurdo, Antonia Valeria Borraccino, Adriana Aloisio, Francesco Russo, Giuseppe Riezzo, Grazia Galeano, Maria Pricci, Bruna Girardi, Francesca Celiberto, Andrea Iannone, Enzo Ierardi, Alfredo Di Leo

**Affiliations:** 1Section of Gastroenterology, Department of Precision and Regenerative Medicine and Ionian Area, University of Bari, 70124 Bari, Italy; giuseppelos@alice.it (G.L.); borraccinovaleria@yahoo.it (A.V.B.); adriana.aloisio95@gmail.com (A.A.); celibertofrancesca@gmail.com (F.C.); ianan@hotmail.it (A.I.); ierardi.enzo@gmail.com (E.I.); 2Functional Gastrointestinal Disorders Research Group, National Institute of Gastroenterology IRCCS “Saverio de Bellis”, 70013 Castellana Grotte, Italy; francesco.russo@irccsdebellis.it (F.R.); giuseppe.riezzo@irccsdebellis.it (G.R.); grazia.galeano@irccsdebellis.it (G.G.); 3THD s.p.a., 42015 Correggio, Italy; mirellapricci@libero.it (M.P.); brunagirardi@virgilio.it (B.G.); 4Ph.D. Course in Organs and Tissues Transplantation and Cellular Therapies, Department of Precision and Regenerative Medicine and Ionian Area, University of Bari, 70124 Bari, Italy

**Keywords:** *Helicobacter pylori*, eradication, concomitant therapy, bismuth quadruple therapy

## Abstract

Concomitant therapy (CT) and bismuth quadruple therapy (BQT) are recommended in geographical areas with high clarithromycin resistance for *Helicobacter pylori* (*H. pylori*) eradication. We compared CT and BQT as the first lines of treatment in a randomized controlled trial. Consecutive patients with H. pylori diagnosed by concordance of both a urea breath test and histology were recruited. For BQT, patients received 3 Pylera^TM^ capsules q.i.d.; for CT, 1000 mg of amoxicillin b.i.d, 500 mg of clarithromycin b.i.d and 500 mg of metronidazole b.i.d. As a proton pump inhibitor, 40 mg of pantoprazole b.i.d was administered. Both regimens lasted 10 days. In total, 46 patients received CT and 38 BQT. Both groups were comparable for age (*p* = 0.27) and sex (*p* = 0.36). We did not record any drop outs; therefore, the intention to treat and per protocol rates coincided. The most common symptoms were heartburn and post-prandial fullness, which were equally present in both groups. The success rate was 95.6% for CT and 100% for BQT (*p* = 0.56). Side effects were recorded in 23.9% and 31.6% of patients in the CT and BQT arms, respectively (*p* = 0.47). The most common ones were abdominal pain (8) and diarrhea (6). In conclusion, CT and BQT are equally effective in our area with high clarithromycin resistance, southern Italy, and showed comparable safety.

## 1. Introduction

Increasing antibiotic resistances are a threat to the eradication of *Helicobacter pylori* (*H. pylori*), as it has been widely demonstrated that they reduce the effectiveness of eradication regimens [1,2,3,4,5]. Southern Italy is a geographical region where the clarithromycin resistance rate exceeds 20% [6]. For this reason, triple therapy effectiveness has decreased and, therefore, cannot be recommended any more [7,8,9,10,11].

The most recent Maastricht guidelines [12] suggest that, in geographical areas with a high clarithromycin resistance rate, the optimal choice for an empirical regimen should be concomitant therapy (CT) or bismuth-based quadruple therapy (BQT). Italian guidelines also suggest sequential therapy as an alternative to CT and BQT [13]. Indeed, the Italian data from the European registry on *Helicobacter pylori* management confirmed that, as first-line treatment, these three strategies show effectiveness of >90% [14].

So far, only one randomized controlled trial has been performed in Italy to compare CT and BQT [15], showing a success rate, respectively, of 95.2% and 85.2%, while most similar randomized controlled trials come from Asian countries [16,17]. A matter of debate in this regard is that BQT may induce more frequently adverse events, as suggested by some evidence [15]. On these bases, in order to provide a further contribution regarding a comparison of the effectiveness and safety of BQT and CT in Italy, we conducted a randomized controlled trial in our area, and we aimed to report the preliminary results of our experience.

## 2. Results

Eighty-four patients with a primary infection were recruited. In total, 46 patients received CT and 38 BQT. The mean age was comparable between groups: 57.0 ± 15.0 for CT and 60.6 ± 13.0 (*p* = 0.27). The two groups were similar with regard to sex (*p* = 0.36), as shown in Table 1. The main comorbidities reported by patients were arterial hypertension (n = 16, 19.0%) and diabetes (n = 8, 9.5%), without any differences between CT and BQT. All patients were recruited because of dyspeptic symptoms, the most common ones being epigastric pain, early satiety/fullness and heartburn, which were reported to have similar levels of prevalence between the two groups. Further details are described in Table 1.

The main endoscopic findings, described in Table 1, were isolated antral hyperemia, which was found in 22 patients, and antral erosive gastritis in 34 subjects. Only two duodenal ulcers were observed. No statistically significant difference between the two groups was found regarding endoscopic pictures.

We did not record any drop-outs; therefore, the intention to treat and per protocol rates coincided. The success rate was 95.6% (95% CI 94.6–96.6) for CT and 100% for BQT (*p* = 0.56). Side effects were recorded in 23.9% and 31.6% of patients in the CT and BQT arms, respectively (*p* = 0.47). The most common ones were abdominal pain (8) and diarrhea (6), as shown in Table 2. Early termination four days before due to adverse events was observed in one patient receiving CT and in another two receiving BQT (*p* = 0.58).

## 3. Discussion

The treatment of *H. pylori* infection is becoming a complex issue worldwide due to the spreading of antimicrobial resistance [18,19,20]. According to this, one of the most important factors influencing the outcome of its treatment is the geographical area because of the different prevalence levels of antimicrobial resistance across different countries [21].

For this reason, clarithromicin-free therapeutic regimens such as bismuth quadruple therapy (BQT), which is not influenced by single clarithromycin or dual clarithromycin and metronidazole resistance, are becoming fundamental not only when used as second-line [22] but also as first-line treatment in geographical areas, including southern Italy, where clarithromycin resistance exceeds 15% [12,13]. Another treatment option to overcome the issue of antibiotic resistance when used as a first-line treatment is concomitant therapy, even if the first option (BQT) tends to be preferred in populations with elevated dual clarithromycin and metronidazole resistance profiles [18,23]. 

In the present randomized controlled trial, according to the preliminary data, we showed that the effectiveness of BQT and CT is similar and their safety is comparable to the empirical first-line treatments of *H. pylori* infection in our area (southern Italy) with high clarithromycin resistance (>20%). In detail, we proved that, in a group of naïve adult patients, 10-day treatment with CT was successful in 95.6%, while the success rate of BQT was 100%. 

Our findings agree with the results of a recent systematic review and meta-analysis of randomized controlled trials by Zagari et al. [24] comparing the efficacy of the two quadruple regimens when used as a first-line treatment for *H. pylori* infection. The review included six studies which were all conducted in Asia [16,17,25,26,27] except for one, which was a multi-center trial from Italy [8]. All the studies involved adult patients with a mean age from 37 to 58 years, with a sample size ranging from 70 to 1080 subjects, who were treated with BQT or CT for 10 [15,17,25,27] or 14 days [16,26]. Similarly to the patients recruited into our trial, most of the subjects complained of dyspeptic symptoms either associated with or without endoscopic findings of peptic ulcers. In all studies, BQT consisted of bismuth subcytrate, tetracycline and metronidazole taken separately, while concomitant therapy consisted of amoxicillin, clarithromycin and metronidazole, with the only exception being in the Italian study [15], which used the “three-in-one” formulation for BQT and tinidazole instead of metronidazole for CT. Among the studies included in the meta-analysis, there was heterogeneity in the choice of PPIs (rabeprazole, lansoprazole, omeprazole, esomeprazole), which were all administered at a standard dose twice daily, except for the study by Kefeli et al. [25], in which, as in our work, a high dose of PPI twice daily was used. The total number of analyzed patients was 1810, of whom 906 received concomitant therapy and 904 standard bismuth quadruple therapy, reaching an overall eradication rate of 87.4% with BQT and 85.2% with CT [15]. 

When analyzing the subgroups in detail, it is interesting to notice that the Asian studies [16,17,25,26,27] showed a small yet significant superiority of BQT over CT (87.5% vs. 84.5%), probably due to the highest prevalence of single and dual clarithromycin and metronidazole resistance in eastern countries [24]. Certainly, similarly to our results, the European study showed comparable success rates between the two regimens [15]. 

The meta-analysis by Zagari et al. [24] reported no difference in eradication rates between BQT and CT at 10 or 14 days. According to this, our choice to administer both treatments for 10 days has been successful, proving that a 10-day regimen is effective, and also considering the benefits in terms of the compliance of the patient, which is an important subject of interest because of its possible influence on treatment outcomes [28,29,30,31,32,33,34]. 

Zagari et al. [24] reported a similar overall incidence of adverse events in patients treated with BQT and CT (52.3% and 48.3%, respectively); the overall rate of early therapy termination due to severe adverse events was 6.5% for BQT and 4.5% for CT. Similarly, our results showed a higher percentage of side effects in patients treated with BQT (31.6%) compared to CT (23.9%); the most common events were abdominal pain, diarrhea and nausea. Early therapy termination was registered in 5.2% of patients receiving BQT and 2.2% of patients receiving CT.

Since the majority of studies on this subject come from Asia, a strength of this trial is that it provides a “head-to-head” comparison between two of the guideline-recommended regimens for the first-line treatment of *H. pylori* infection in a geographical area with high clarithromycin resistance, fulfilling the demand for new evidence in western countries. 

Whether 14-day regimens in an empiric setting should be preferred to 10-day regimens is a matter of debate. Guidelines suggest a prolonged regimen. Despite recent advancements, however, the clear superiority of prolonging the duration to 14 days has not been demonstrated [35]. A small increase in therapy success certainly is often counterbalanced by more common adverse effects. Furthermore, bismuth salts are an effective weapon against *H. pylori*, as they have an intrinsic antibacterial effect, and they do not elicit any antimicrobial resistance. It has been demonstrated that adding bismuth to an eradication regimen may result in an additional 30%–40% gain in the success with resistant infections [36]. This may explain the slightly better efficacy of BQT over CT in our study. Therefore, when available, adding bismuth should always be considered when facing difficulties in eradicating *H. pylori.*

Nevertheless, there are some limitations, such as the limited sample size and the non-availability of *H. pylori* antimicrobial susceptibility profiles. Liou et al. [17], whose multi-center study was included in the meta-analysis by Zagari et al. [24], provided eradication rates by antibiotic resistance, demonstrating that, in terms of efficacy, BQT was superior to CT in subjects either with single (success 89% versus 72%) or dual clarithromycin and metronidazole resistance (94% versus 59%). 

In the near future, a shared approach, with concentration on clinical cases in a few expert centers, will be of paramount importance to ensure quality in *H. pylori* eradication, as already confirmed by evidence of similar experiences in other fields [37]. Furthermore, it is appropriate that clinical trials for *H. pylori* treatment will be supported by resistance information; this approach will be pivotal in order to reach, whenever possible, a basis for the formulation of a tailored therapy taking into account individual antimicrobial resistance profiles [38,39,40,41,42,43].

## 4. Materials and Methods

Consecutive patients complaining of dyspeptic symptoms [44,45] with *H. pylori* diagnosed by concordance of both the urea breath test and histology were recruited. The study was conducted in agreement with the indications of the Declaration of Helsinki, and the local Ethics Committee approved the protocol (Azienda Ospedaliera Universitaria Consorziale Policlinico di Bari, protocol n. 74413). All participants signed written informed consent.

All patients were naïve and underwent urea breath tests (UBTs) before enrollment (Breath quality-UBT 75 mg/10 mL-13C Urea). A baseline breath sample was gathered, asking the patient, who had been fasting for at least 6 hours, to blow inside of a test tube through a straw. Then, the patient was given an oral solution of 75 mg of ^13^C urea diluted in 200 mL of water (breath quality—UBT 75 mg/10 mL–13C urea). After 30 minutes, a second breath sample was collected. The exhaled air was analyzed using a mass spectrometer, allowing us to measure the amount of ^13^CO_2_ compared to the total exhaled CO_2_. Positivity was defined as a Delta over baseline (DOB) > 4‰ [46,47,48].

Esophagogastroduodenoscopy was performed in all patients with the execution of two antrum and two body biopsy samples, oriented by [49,50,51]. Then, the samples were embedded in formalin and analyzed by a dedicated pathologist, using hematoxylin eosin and Giemsa staining for the identification of *H. pylori*.

A computer-generated sequence was used for randomization. For BQT, patients received 3 Pylera^TM^ capsules q.i.d.; for CT, 1000 mg of amoxicillin b.i.d, 500 mg of clarithromycin b.i.d and 500 mg of metronidazole b.i.d. As a proton pump inhibitor, 40 mg of pantoprazole b.i.d was administered [12,52]. Both regimens lasted 10 days. Six to eight weeks after the end of the therapy, eradication was verified by a urea breath test, and side effects were recorded. Compliance with the eradication regimen was defined as consumption of >80% of the medication. Eradication rates were expressed both as intention to treat and per protocol. 

Discrete variables, expressed as proportions/percentages, were analyzed by Fisher’s test or χ^2^ test with Yates correction when required. Student’s test was used for continuous variables, reported as the mean ± standard deviation. All analyses were conducted with two-tailed tests, with statistical significance at *p* < 0.05. Analyses were conducted using GraphPad Prism statistical software version 5 for Windows, San Diego, CA, USA.

## 5. Conclusions

In conclusion, this randomized controlled trial confirmed that CT and BQT are equally effective, even in our area with high clarithromycin resistance, southern Italy, and showed comparable safety.

## Figures and Tables

**Table 1 antibiotics-13-00348-t001:** Demographic characteristics, symptoms and endoscopic pictures of recruited patients.

	Concomitant Therapy (n = 46)	Bismuth Quadruple Therapy (n = 38)	*p*
Sex M/F	14/32	16/22	0.36
Age (mean ± standard deviation and range)	57.0 ± 15.0 (25–84)	60.6 ± 13.0 (24–79)	0.27
Main comorbidities, n (%)	10 (21.7%)	9 (23.6%)	1
Early satiety, n (%)	6 (13.0%)	2 (5.2%)	0.28
Epigastric pain, n (%)	12 (26.1%)	16 (42.1%)	0.16
Heartburn, n (%)	16 (34.8%)	14 (36.8%)	1
Post-prandial fullness, n (%)	9 (19.6%)	12 (31.6%)	0.21
Endoscopic picture, n (%) - Antral hyperemia - Antral erosive gastritis - Gastric ulcer - Duodenal ulcer - Normal	14 (30.4%) 18 (39.1%) 0 (0%) 2 (4.3%) 12 (26.2%)	8 (21.0%) 16 (42.1%) 0 (0%) 0 (0%) 14 (36.9%)	0.35

**Table 2 antibiotics-13-00348-t002:** Success rate and adverse events of BQT and CT.

	Concomitant Therapy (n = 46)	Bismuth Quadruple Therapy (n = 38)	*p*
Success rate	44 (95.6%)	38 (100%)	0.56
Adverse events	11 (23.9%) 4 abdominal pain 2 diarrhea 1 weakness 1 bloating 1 nausea 2 other	12 (31.6%) 4 diarrhea 2 weakness 2 bloating 4 nausea	0.47
Early therapy termination	1 (2.2%)	2 (5.2%)	0.58

## Data Availability

The data are contained within the article.

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
