# Peer review of "Concomitant and Bismuth Quadruple Therapy for Helicobacter pylori Eradication in Southern Italy: Preliminary Data from a Randomized Clinical Trial"

_antibiotics, 2024, doi:10.3390/antibiotics13040348_

Round 1

Reviewer 1 Report

Comments and Suggestions for Authors

1. Why did the authors choose a 10-days therapy for H.pylori infection?

2. Would 14 days therapy give better results for patients on concomitant therapy?

3. The authors should emphasize whether this was a primary infection with H.pylori in all patients included in the study?

Author Response

  1. Why did the authors choose a 10-days therapy for H.pylori infection?

We decided to use 10-day regimens as, at the time when the protocol was approved by the ethic committee, guidelines recommended a 10-days regimens in first line. Despite recent advancement, however, a clear superiority of prolonging duration to 14 days has not been demonstrated (Yang EH, Chen WY, Chiang HC, Li CH, Wu IH, Chen PJ, Wu CT, Tsai YC, Cheng WC, Huang CJ, Sheu BS, Cheng HC. 10-Day versus 14-day bismuth quadruple therapy for first-line eradication of Helicobacter pylori infection: a randomised, open-label, non-inferiority trial. EClinicalMedicine. 2024;70:102529). A small increase of therapy success, indeed, is often counterbalanced by a rise of adverse effects.

  1. Would 14 days therapy give better results for patients on concomitant therapy?

As discussed in the previous answer, there is no clear evidence that a 14-day therapy would give better results than a 10-fay one. On the other hand, a 95.6% success rate for 10-day concomitant regimen is by now very high.

  1. The authors should emphasize whether this was a primary infection with H. pylori in all patients included in the study?

We confirm that all patients had a primary infection. This was better stated in revised text.

Reviewer 2 Report

Comments and Suggestions for Authors

This was a randomized controlled trial on the efficacy and safety of concomitant therapy and bismuth quadruple therapy for H. pylori eradication in Southern Italy . A total of 84 patients were enrolled, and the PP analysis showed a eradication success rate of 95.6% for CT and 100% for BQT, without significant differences. The authors emphasized the efficacy and safety of both regimens in Southern Italy. 

However, there were some serious concerns during the review process.

#. Several p-values are different in the manuscript, table, and abstract.

#. I don't understand why the authors did not record dropouts to analyze the ITT and PP results. Since non-compliance could be due to side effects of medication or complex medication regimen, dropout is an important issue, which should be considered in the analyses.

Comments on the Quality of English Language

The English of the manuscript needs improvement.

Author Response

This was a randomized controlled trial on the efficacy and safety of concomitant therapy and bismuth quadruple therapy for H. pylori eradication in Southern Italy. A total of 84 patients were enrolled, and the PP analysis showed a eradication success rate of 95.6% for CT and 100% for BQT, without significant differences. The authors emphasized the efficacy and safety of both regimens in Southern Italy.

However, there were some serious concerns during the review process.

#. Several p-values are different in the manuscript, table, and abstract.

We thank the reviewer for raising up this point. Wrong values were reported in the abstract and they have been corrected.

#. I don't understand why the authors did not record dropouts to analyze the ITT and PP results. Since non-compliance could be due to side effects of medication or complex medication regimen, dropout is an important issue, which should be considered in the analyses.

As we remarked in page 2, “We did not record any drop-out, therefore intention to treat (ITT) and per protocol (PP) rated coincided.”

Reviewer 3 Report

Comments and Suggestions for Authors

This brief report demonstrates alternatives to traditional H. pylori treatments. Minor concerns exist, but none that the authors can not easily fix.

- Title - Itlaized Helicobacter pylori
- Table 1 and 2. Move down (n = ) for clarity
- Cut back on abbreviations. It makes it very difficult to read.

Comments on the Quality of English Language

A read-through would be helpful, but no major grammatical errors were found.

Author Response

This brief report demonstrates alternatives to traditional H. pylori treatments. Minor concerns exist, but none that the authors cannot easily fix.

- Title - Itlaized Helicobacter pylori

We performed the requested change.

- Table 1 and 2. Move down (n = ) for clarity

We performed the requested change.

- Cut back on abbreviations. It makes it very difficult to read.

We left only CT, BQT and PPI as abbreviations, since they are commonly used worldwide.

Reviewer 4 Report

Comments and Suggestions for Authors

Very pleasant paper to read, we still talk about Helycobacter Pilory, which we know is strongly implicated in the onset of gastric neoplasia, especially in a region, southern Italy, where it is also found at a young age. I really like the approach given by our fellow researchers and the second line of treatment, if we can call it that, in case of positivity, once the presence of this bacterium in the stomach is found, with resistance to clarithromycin. Personally, I would prefer the results to the discussion which, together with the conclusions, represents a worthy condensing of the work and of the theses set out in the introduction. I like to suggest the inclusion in the bibliography of a work (doi.org/10.3390/jcm12072708 to be cited) which apparently does not seem to be directly related to the study in question, but which means that the concentration of pathologies in a few centers increases their experience, knowledge and consequently improves the outcome of the treatment; considering that gastroenterologists and surgeons work side by side on a daily basis and the conjugation of mutual experience with common interests certainly improves through harmony, the result. Good English, good iconography, the bibliography supports theses and conclusions, theses and conclusions in line

Comments on the Quality of English Language

English needs to be slightly improved

Author Response

Very pleasant paper to read, we still talk about Helycobacter Pilory, which we know is strongly implicated in the onset of gastric neoplasia, especially in a region, southern Italy, where it is also found at a young age. I really like the approach given by our fellow researchers and the second line of treatment, if we can call it that, in case of positivity, once the presence of this bacterium in the stomach is found, with resistance to clarithromycin. Personally, I would prefer the results to the discussion which, together with the conclusions, represents a worthy condensing of the work and of the theses set out in the introduction. I like to suggest the inclusion in the bibliography of a work (doi.org/10.3390/jcm12072708 to be cited) which apparently does not seem to be directly related to the study in question, but which means that the concentration of pathologies in a few centers increases their experience, knowledge and consequently improves the outcome of the treatment; considering that gastroenterologists and surgeons work side by side on a daily basis and the conjugation of mutual experience with common interests certainly improves through harmony, the result. Good English, good iconography, the bibliography supports theses and conclusions, theses and conclusions in line

We thank the reviewer for the kind appreciation of our article. We added the suggested article and discussed it in the revised manuscript.

Round 2

Reviewer 2 Report

Comments and Suggestions for Authors

I am afraid that the authors did not show adequate or sincere response to the raised questions.

Comments on the Quality of English Language

The English needs to be improved.

Author Response

Please precise the point you were in doubt.

The difference in percentages have been corrected in the Abstract.
As we said, we did not record drop outs. Early termination did not count in  our analysis as drop out, and results of eradication after therapies in these cases were recorded and discussed.